# KARMA: Keyword-Aware Representation Modification for Efficient and Robust Model Amnesiac Unlearning

## Abstract

Pre-trained Language Models (LMs) struggle with efficiently removing specific data samples and associated knowledge due to their massive scale and computational requirement. Existing machine unlearning methods suffer from excessive parameter updates and an imbalanced forgetting-remaining performance. We first derived critical insight that fine-tuning only a single layer of the model achieves competitive performance to full-model fine-tuning. Inspired by this observation, we introduce KARMA (Keyword-Aware Representation Modification for Model Amnesiac Unlearning), which efficiently forgets representation traces by selectively perturbing the embedding parameters of semantically critical tokens, while restricting parameter updates within a bounded spherical region to preserve stability. Specifically, to identify high-influence keywords, we first introduce a Fisher scoring mechanism that precisely captures the semantics of data to be forgotten. To further enhance privacy during the unlearning process, we propose a keyword-driven pseudo sample based method that eliminates the need for raw data by inserting keyword embeddings within irrelevant corpora. Moreover, to mitigate the adverse impact on the remaining samples, we propose a bounded fine-tuning regularization strategy to prevent excessive semantic drift in the representation space. The efficiency of KARMA is underpinned by rigorous convergence radius analysis, and the robustness of KARMA on remaining samples is theoretically proved by the bounded regularization strategy. Experiments on sentiment classification show that KARMA achieves near-retraining efficacy with a 99.5% reduction in parameter updates compared to gradient-based methods, while exhibiting a low performance degradation on retained data. Codes are available at https://anonymous.4open.science/r/KARMA-4501.

## 1 Introduction

Pre-trained language models (LMs), trained on large-scale corpora, have become fundamental tools in natural language processing to accelerate downstream applications. However, this paradigm presents challenges when users need to remove specific samples and the associated knowledge, especially when increasing concerns are given over data privacy and compliance with regulations like GDPR Voigt & von dem Bussche (2017). As a promising solution, recent works in machine unlearning have explored multiple strategies tailored to language models. Some methods estimate the influence of training samples on model outputs using influence functions to guide the gradient update direction Wang et al. (2023b). Others design custom loss functions, such as asymmetric Kullback–Leibler (KL) divergence or negative gradient alignment, to selectively forget specific knowledge Wang et al. (2025a); Yao et al. (2024). However, these two types of methods share a critical limitation: they focus on "how to forget" but overlook "how to forget efficiently," laying the groundwork for excessive computational costs in large-scale models.

Despite their contributions, existing unlearning methods remain impractical for resource-constrained scenarios. Firstly, they require massive parameter updates and face scalability issues, since updated parameters grow with model size. For example, computing a full Hessian requires $O(P^2)$ memory, even approximations like the Fisher matrix remain costly. Moreover, current approaches lack fine-grained control over update regions, causing unnecessary perturbations to retained samples. Strong

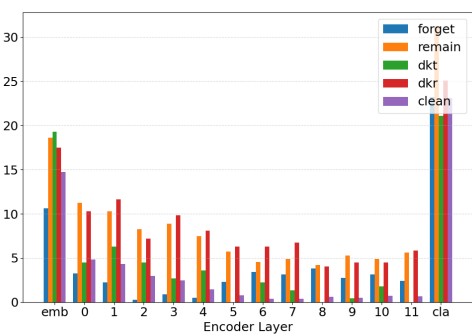 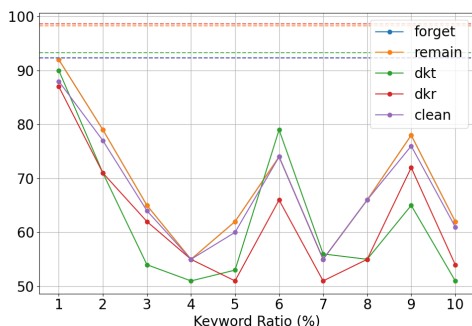

Figure 1: Performance of retraining with fine-tuning different layers. The x-axis represents the layers that remain unfrozen during fine-tuning, while the y-axis denotes the absolute percentage difference from the retrain performance across different metrics.

Figure 2: Impact of keyword ratio on accuracy in unlearning. The x-axis indicates the percentage of keyword length relative to the sentence length, while the y-axis shows the test performance across different metrics. The horizontal line represents the retrain performance.

updates erase target knowledge but degrade performance on remaining data, while conservative updates preserve utility yet yield incomplete forgetting.

While parameter-efficient fine-tuning methods (e.g., LoRA, SparseGrad) reduce training cost by restricting updates to a small subset of parameters, they are not well-suited for unlearning. These methods typically operate on fixed parameter subsets, which can cause excessive model changes even when forgetting only a few samples. Moreover, they lack semantic awareness, making them less applicable in privacy-sensitive scenarios mentioned above. To enable forgetting within a more targeted parameter space, we study how different layers contribute to knowledge retention during unlearning process. In the setting of layer-wise fine-tuning unlearning, we use full-parameter fine-tuning as the baseline and measure the performance deviations across five evaluation metrics (see Section 5 for more details) when only a single layer is fine-tuned. As shown in Figure 1, fine-tuning individual layers have performance comparable on the forget set to full fine-tuning. However, fine-tuning the embedding layer or the classifier leads to noticeable performance drops on the retain set, due to interference with unrelated parameters. To further reduce the number of parameters used during unlearning, we explore partial fine-tuning within a single layer. Since only the embedding layer has clear semantic representation, we focus on selectively fine-tuning in this layer to avoid unnecessary disturbance and enable more targeted forgetting. As shown in Figure 2, selecting only 1% of the most relevant tokens per sample achieves the best unlearning performance. Carefully chosen keywords effectively capture the semantics of individual samples, whereas using too many tokens can lead to negative interference with irrelevant samples. As show in the above study, fine-tuning critical tokens within specific layers can effectively achieve unlearning. Leveraging semantic information helps confine the impact of unlearning to a targeted scope, thereby minimizing catastrophic forgetting effect.

Build on above observations, we propose KARMA (Keyword-Aware Representation Modification for Efficient and Robust Model Amnesiac Unlearning), a lightweight framework for targeted unlearning in pre-trained language models. KARMA identifies semantically critical tokens via a Fisher scoring–based mechanism and fine-tunes only their embedding parameters to remove associated knowledge, ensuring minimal parameter updates and preserving overall model behavior. To further mitigate unintended side effects on retained samples, we propose a bounded fine-tuning regularization to constrain semantic drift in the representation space. Theoretical analysis supports our design, where convergence radius analysis confirms its efficiency, and the robustness of retained knowledge is guaranteed by the bounded regularization strategy. For privacy-sensitive settings, we further introduce a key-aware-only pseudo sample construction approach, enabling unlearning without using raw data or full parameters.

Our contributions are as follows:

- We propose KARMA, a novel machine unlearning method based on layer-restricted embedding fine-tuning, which is both theoretically grounded and empirically motivated. By

confining updates to a single layer and a bounded parameter space, KARMA achieves effective unlearning with significantly reduced parameter updates, while maintaining high performance on retained data.

- We introduce a Fisher scoring–based semantic keyword extraction mechanism to identify the most influential tokens in the samples to be forgotten. This enables targeted fine-tuning at the keyword level, drastically reducing the number of parameters to be modified. Furthermore, we develop a privacy-preserving pseudo sample strategy that supports unlearning under limited-information settings, without access to raw training samples.

- We theoretically analyze the fine-tuning process under a Lipschitz continuity constraint, and provide a convergence radius analysis to justify the stability and bounded impact of representation updates. These theoretical insights are validated through experiments on sentiment classification tasks, demonstrating comparable unlearning efficacy to retraining with up to 60% fewer updates, and strong resistance to membership inference attacks.

## 2    RELATED WORK

Previous research on unlearning in language models primarily focuses on precision Wang et al. (2025b; 2023b), robustness Yuan et al. (2025), and functionality Yuan et al. (2025). To adapt PLMs efficiently, various parameter-efficient fine-tuning (PEFT) methods have been proposed. Adapter tuning Houlsby et al. (2019), LoRA Hu et al. (2022), and partial update method Zaken et al. (2022) enable task-specific adaptation by updating only a small subset of parameters. These techniques offer scalability and modularity, making them attractive for scenarios requiring lightweight model updates. Recent works have explored integrating PEFT into unlearning to reduce retraining costs. For example, adapter-based methods isolate forgettable knowledge without altering the full model, while LoRA variants apply low-rank masking to suppress sensitive information. These approaches demonstrate the potential of PEFT in facilitating efficient and targeted unlearning. Hu et al. propose a novel parameter-effect module-based unlearning method to provide detoxification by generating effective fabricated content Hu et al. (2024). Cha et al. Cha et al. (2025) propose an efficient unlearning method that combines Inverted Hinge Loss with LoRA layers whose initialization is adaptively adjusted using Fisher Information, effectively suppressing sensitive information in generative models. However, the method relies on specific architectures and lacks aware of semantic granularity, limiting its applicability in low-resource or interpretability-sensitive scenarios.

Despite their efficiency, previous methods often tune numerous parameters and ignore semantic representation, leading to suboptimal forgetting or privacy leakage. To enhance unlearning specificity and efficiency, our proposed KARMA constrains both the scope and value range of fine-tuned parameters through a lightweight regularization mechanism. To further support forgetting under limited data access, KARMA introduces a pseudo sample construction method guided by semantic cues, enabling effective unlearning without reliance on original data.

## 3    METHODOLOGY

### 3.1    OVERVIEW

In this section, we present the proposed KARMA framework, which achieves effective unlearning by modifying only a small subset of highly influential embedding parameters in the model. KARMA consists of two components: (1) a **Keywords Selection** module, and (2) a **Selective Representation Modification** module, which supports both full-sample and keyword-aware-only training modes. Specifically, to enable forgetting with minimal parameter updates, we first employ the Keywords Selection module to identify keywords for each individual sample in the subset to be forgotten, and then leverage the Selective Representation Modification module to fine-tune only the embeddings of these selected keywords, thereby accomplishing efficient unlearning.

### 3.2    PROBLEM FORMULATION.

Let $\mathcal{D} = \{(x_i, y_i)\}_{i=1}^{N}$ be the training dataset, where $N$ is the number of samples and $y_i$ is the label of sample $x_i$. We fine-tune a pre-trained model $f(\cdot; \theta_p)$ on $\mathcal{D}$ to obtain a downstream model $f(\cdot; \theta)$,

where $\theta_p$ and $\theta$ denote the model parameters before and after fine-tuning, respectively. Let $\mathcal{D}_f \subset \mathcal{D}$ be the subset to be forgotten, while $\mathcal{D}_r := \mathcal{D} \setminus \mathcal{D}_f$ be the remaining dataset. Performing unlearning is to erase any influence of $\mathcal{D}_f$ from $\theta$ by obtaining a set of new parameters $\hat{\theta}$ that closely align with the performance of the retrained model parameters $\theta_r$, where $\theta_r$ is fine-tuned from scratch by the $\theta_p$ on $\mathcal{D}_r$. Given that retrained models typically exhibit performance degradation on $\mathcal{D}_f$ (performing similarly to non-fine-tuned models) while retaining nearly full performance on $\mathcal{D}_r$, KARMA aims to eliminate information related to $\mathcal{D}_f$ while maintaining performance on $\mathcal{D}_r$ comparable to retrained models. To minimize parameter modifications and computational overhead, KARMA selectively fine-tuning the embeddings of a small set of influential tokens instead of retraining the full model. In addition, KARMA supports privacy-sensitive scenarios like remote unlearning requests, by enabling forgetting of $\mathcal{D}_f$ using only selective keywords without exposing the full original data. To accurately forgetting the subset to be forgotten $\mathcal{D}_f$ and preserving performance on the remaining subset $\mathcal{D}_r$, excessive semantic drift is relieved by a bounded fine-tuning regularization strategy (see Equation 5), which is theoretically guaranteed in Theorem 1.

### 3.3 KEYWORDS SELECTION

In the keyword selection module, we adopt a Fisher-scoring method to assign scores to the subword tokens of each instance. These subword tokens are then restored into natural language words, and the words are ranked according to the average score of their constituent subword tokens. Finally, the highest-scoring words are selected as keywords for the instance.

Specifically, the score of each token can be calculated as follows:

$$\text{score}(t) = \left\| g_t \odot \frac{1}{g_t^2} \right\|_2, \tag{1}$$

where $g_t = \nabla_{e_t} \mathcal{L}$ denotes the gradient of the loss $\mathcal{L}$ with respect to the embedding vector $e_t$ of token $t$. Here, $g_t^2$ provides a diagonal approximation of the Fisher Information Matrix. The score reflects the sensitivity of the model to perturbations in token $t$. After scoring subword tokens, we compute the average score for each consecutive sequence of $n$ tokens. The highest-scoring $n$-gram is then reconstructed into natural words by concatenating subwords back into complete words or phrases via the tokenizer's standard detokenization procedure (e.g., 'play' + '##ing' $\rightarrow$ 'playing'). This mapping ensures that the extracted instance keywords $k_i$ are interpretable and aligned with human-readable language. Notably, it carries an additional implication for the key-aware-only mode (detailed in Section 3.4): when constructing pseudo-samples, we can avoid inserting raw subword pieces into the text and instead use natural language expressions. As a result, the semantic readability and coherence are preserved with these natural language expressions while preventing the exposure of raw subword fragments.

### 3.4 SELECTIVE REPRESENTATION MODIFICATION.

To enable sample-level machine unlearning by altering the semantic representation of samples in $\mathcal{D}_f$, we adopt a sample confusion strategy. We resample each $y_i \in D_f$ to obtain the corresponding confusing sample label $\hat{y}_i = resample(y_i)$, where $resample(\cdot)$ chooses $\hat{y}_i \in Y$ following the distribution of $\{y\}$. $Y$ is the label set of $\mathcal{D}$. By replacing the original labels with confusing alternatives, we weaken the association between input texts and their ground-truth labels. At the same time, we guide the model's predictions on these samples to align more closely with those of the original pre-trained model.

To accommodate varying privacy requirements, we propose two unlearning strategies: full-sample mode and key-aware-only mode. In the full-sample mode, we freeze all parameters except for the embedding layer $\theta^e$ and fine-tune the model with $\mathcal{D}_f$. Specifically, we fine-tune only the parameters $\theta^{e^{\mathcal{K}}}$ associated with the keyword set $\mathcal{K} = \{k_i | i \in N\}$ extracted from $\mathcal{D}_f$, using the confusing labels $\{\hat{y}\}$. The unlearned model can be represented by the updated embedding parameters $\hat{\theta}_e^{\mathcal{K}}$, which are obtained by optimizing the following objective:

$$\theta^{\hat{e}^{\mathcal{K}}} := \theta^{e^{\mathcal{K}}} - lr * (\alpha \mathcal{L}_{CE} + (1 - \alpha)\mathcal{L}_{KL}). \tag{2}$$

Here, $lr$ is the learning rate, $\alpha$ balances the contribution of $\mathcal{L}_{CE}$ and $\mathcal{L}_{KL}$. The details of $\mathcal{L}_{CE}$ can be described as follows:

$$\mathcal{L}_{CE} = \mathbb{E}(\hat{y}, f(x, \theta^{e^{\mathcal{K}}})), \tag{3}$$

where $\mathbb{E}(\hat{y}, f(x, \theta^{e^{\mathcal{K}}}))$ is the cross-entropy loss. The details of $\mathcal{L}_{KL}$ can be described as follows:

$$\mathcal{L}_{KL} = KL(f(x, \theta^{e^{\mathcal{K}}}); f(x, \theta_p^{e^{\mathcal{K}}})), \tag{4}$$

where $KL$ is the KL divergence which measures the distance between local model $\theta^{e^{\mathcal{K}}}$ and original pre-trained model $\theta_p^{e^{\mathcal{K}}}$. The output of $\theta^{e^{\mathcal{K}}}$ is the prediction result of the unlearning model, and the output of $\theta_p^{e^{\mathcal{K}}}$ can be regarded as the prediction if the pre-trained model has not learned any knowledge of $\mathcal{D}_f$. By minimizing the difference between the outputs of $\theta^{e^{\mathcal{K}}}$ and $\theta_p^{e^{\mathcal{K}}}$, we can approximate the performance of the local model to the performance of the model that has not been fine-tuned on $\mathcal{D}$.

To stabilize training and avoid excessive semantic drift, we constrain each updated embedding vector in terms of magnitude. Specifically, we rescale each updated keyword embedding $\hat{\theta}^{e^{k_i}}$ to match the original norm:

$$\hat{\theta}^{e^{k_i}} \leftarrow \hat{\theta}^{e^{k_i}} \cdot (\|\theta^{e^{k_i}}\|_2 / \|\hat{\theta}^{e^{k_i}}\|_2) \tag{5}$$

This operation preserves the original vector magnitude and regularizes the update, preventing embeddings from over drifting in representation space. By bounding the scale of modifications, it acts as a constraint that helps stabilize the unlearning process and maintain consistency with the original representation distribution, as further analyzed in Section 4.

The unlearning process in the key-aware-only mode closely follows that of the full-sample mode. The key difference is that the key-aware-only mode uses only pseudo samples set $\mathcal{D}^*$ constructed by inserting selected keywords instead of $\mathcal{D}_f$. Specifically, for each $k_i \in \mathcal{K}$, we extract a segment from an irrelevant corpus with larger sentence spaces, whose length matches the average sample length in $D_f$, and randomly insert $k_i$ into the segment. For instance, to unlearn instances from movie reviews, we can use WikiText-103 as the irrelevant corpus. The confusing label associated with the instance containing $k_i$ is then assigned to this pseudo-sample. These pseudo-samples are then used to fine-tune $\theta_e$ for unlearning, without exposing the original content of $D_f$.

## 4 THEORETICAL ANALYSIS

In this section, we present the robustness and convergence analysis with the following assumptions:

**Assumption 1:** (Smoothness). $L$ is $\ell - smooth$ if $\forall x, y \in \Re^d$:

$$L(x) - L(y) + (x - y)^T \bigtriangledown L(x) \leq \frac{\ell}{2}\|x - y\|_2^2. \tag{6}$$

**Assumption 2:** (Bound of Variance). Let $\tilde{\triangle}$ be the variance of the weighted matrix. According to equation 10 and the Lipschitz continuous, the weighted matrix is bounded: $\|W + \tilde{\triangle}\|_2^2 = \|W\|_2^2$.
**Assumption 3:** Any gradient has a uniform upper bound, i.e. $\exists G > 0, \forall i, s.t. \|\bigtriangledown L_t\| < G$.

**Theorem 1.** *KARMA has a bigger Certified Radius for the remaining learning task.*

***Proof.*** For the retraining method, under the assumption that each layer of the network is $\ell - smooth$, the global perturbation propagation satisfies the following equation,

$$\|\tilde{\triangle}_K\|_2 = \prod_{k=1}^{K} \|W_l\|_2 \cdot L_\delta \cdot \delta \tag{7}$$

where $W_l$ is the weighted matrix of the $l$-th layer, $L_\delta$ is the Lipschitz constant of $\delta$.

Considering the acceptable perturbation threshold as $\eta$, we can get the Certified Radius for the retraining method as folllows,

$$\epsilon_{robust}^{retrain} = \frac{\eta}{\prod_{k=1}^{K} \|W_l\|_2 \cdot L_\delta \cdot \delta} \tag{8}$$

For the embedding layer finetune method without constraint (equation 10), under the assumption that the embedding layer is $\ell - smooth$, the robustness for the remaining samples $x$ should satisfy $||\tilde{\triangle}x||_2 \leq \eta$.

Suppose $x_{//}$ is the projection in the space of column $W$, i.e., $x_{//} = Ww^+x$, where $W^+$ is the pseudo-inverse of $W$. Suppose $x_\perp$ is the projection in the orthogonal complementary space of $W$, i.e., $x_\perp = x - x_{//}$. Then, the influence of parameter perturbation on the output is $\tilde{\triangle}x = \tilde{\triangle}x_{//} + \tilde{\triangle}x_\perp$.

For $||\tilde{\triangle}x_{//}||_2$, according to Cauchy-Schwarz inequality, we can get $||\tilde{\triangle}x_{//}||_2 \leq ||\tilde{\triangle}||_2||x_{//}||_2$. Further considering the parameter perturbation $||\tilde{\triangle}|| = \epsilon_{robust}^{E-non}$, we can get $||\tilde{\triangle}x_{//}||_2 \leq \epsilon_{robust}^{E-non} \frac{||wx_{//}||_2}{||w||_2}$.

For $||\tilde{\triangle}x_\perp||_2$, since without constraint by $w$, we can get $||\tilde{\triangle}x_\perp||_2 \leq \epsilon_{robust}^{E-non}||x_\perp||_2$.

Then, we can get the Certified Radius for the embedding layer finetune method without constraint (equation 10), i.e., $\epsilon_{robust}^{E-non} = \frac{\eta}{\frac{||wx_{//}||_2}{||w||_2}+||x_\perp||_2}$.

According to Cauchy-Schwarz inequality and Expansion, we can get the low bond of $\epsilon_{robust}^{E-non}$ as follows,

$$\epsilon_{robust}^{E-non} \geq \frac{\eta}{||x_{//}||_2 + ||x_\perp||_2} \tag{9}$$

For the embedding layer finetune method with constraint equation 10 (i.e., KARMA), according to assumption 2, the relation can be approximated as $w^T\tilde{\triangle} = 0$. Then, we can get $||\tilde{\triangle}x_{//}||_2 = 0$. The Certified Radius for KARMA is $\epsilon_{robust}^{KARMA} = \frac{\eta}{||x_\perp||_2}$.

From the above analysis process, we can obtain the inequality $\epsilon_{robust}^{KARMA} \geq \epsilon_{robust}^{E-non} > \epsilon_{robust}^{retrain}$, which means that our method KARMA is more robust for the remaining learning tasks while fine-tuning for unleaning task.

**Theorem 2.** *Consider $\theta$ to be a a variable constant. Then we have the covergence radius for KARMA:*

$$R_c = \frac{\eta_t G}{2} \tag{10}$$

***Proof.*** The parameter update rule for KARMA:

$$\theta_{t+1} = \theta_t - \eta_t(\bigtriangledown L(\theta_t)) \tag{11}$$

Then, we can get

$$||\theta_{t+1}||^2 = ||\theta_t||^2 - 2\eta_t < \theta_t, \bigtriangledown L(\theta_t) > +\eta_t^2||\bigtriangledown L(\theta_t)||^2 \tag{12}$$

We need to show that $||\theta_{t+1}||^2 \leq R^2$ under the assumption that $||\theta_t|| \leq R$. According to Cauchy-Schwarz inequality, $< \theta_t, \bigtriangledown L(\theta_t) > \geq GR$. Then, we can get the inequality $R^2 + 2\eta_t GR + \eta_t^2 R^2 \leq R^2$.

The above formulation is simplified as $2\eta_t GR + \eta_t^2 R^2 \leq 0$. We can get $R \geq R_c = \frac{\eta_t G}{2}$.

## 5 EXPERIMENTS

### 5.1 DATASETS.

We utilize semantic analysis datasets, IMDb Tripathi et al. (2020) and SST-2 Srinivasan et al. (2016), to demonstrate the effectiveness of the proposed method. To observe the impact of unlearning on the model's performance with respect to both the forgotten and the remaining samples, $\beta\%$ of the data is randomly selected as the forgotten samples, with the rest serving as the remaining samples. Here, we set $\beta = 1\%$ by default.

## 5.2 MODELS AND BASELINES.

In this paper, the proposed method and three baselines are implemented based on the bert-base-uncased model. For KARMA, we set the learning rate to 2e-2 and alpha to 0.8, with batch sizes of 32 for the IMDb dataset and 1024 for the SST-2 dataset. In addition, KARMA performs only a single epoch of fine-tuning across all settings.

The three baselines are "gold standard" model retraining (Retrain), fine-tuning with a high learning rate (Fine-tune), gradient ascent (GA), and KGA Wang et al. (2023a). All the baselines use Adam as the optimizer and the batch sizes are set as the proposed method. Detailed implementation can be found in Section 9.1.

## 5.3 EVALUATION METRICS.

The most intuitive impact of unlearning on classification models is the overall classification accuracy. Therefore, we primarily use inference accuracy for performance analysis, and additionally report F1-score to better capture unlearning effects on remained data. We first measure the accuracy of the local model and the model after the unlearning task on the test dataset $\mathcal{D}_t$ (including both IMDb and SST-2 datasets). To quantify the effectiveness of forgetting the data to be forgotten, we also test the inference accuracy of the unlearned model on $\mathcal{D}_f$. Then, we apply the k-means algorithm to the remaining samples and test samples separately, to obtain the semantic closest subset $\mathcal{D}_{kr}$ and $\mathcal{D}_{kt}$, respectively. The performance metrics on these two datasets, which contain samples from $\mathcal{D}_r$ and $\mathcal{D}_t$ that are semantically similar to $\mathcal{D}_f$, quantify the effectiveness of the unlearning method in removing semantic knowledge. Specifically, degraded model performance on these subsets correlates with the extent of semantic knowledge erasure, where poorer performance indicates more comprehensive removal of target knowledge.

We further evaluate the privacy protection capability of our method via Membership Inference Attack (MIA) tests using two metrics, $\Delta$ASR and $\Delta$AUC. $\Delta$ASR measures the change in Attack Success Rate (ASR) before and after unlearning, smaller values indicate lower risk of information leakage. $\Delta$AUC quantifies the deviation of the model's Area Under the Curve (AUC) from 50% after unlearning, smaller value implies that an adversary's classifier is less effective.

## 5.4 RESULTS AND ANALYSIS.

**Overall performance.**

Table 1 shows the results of the sentiment analysis task for unlearning the original model in different Metrics. In this experiment, we set 3 cluster centers and 1 nearest neighbor sample to observe the performance of $\mathcal{D}_{kr}$ and $\mathcal{D}_{kt}$. We summarize the performance of different unlearning methods across multiple evaluation datasets ($\mathcal{D}_r$, $\mathcal{D}_f$, $\mathcal{D}_t$, $\mathcal{D}_{kr}$, and $\mathcal{D}_{kt}$) on IMDb and SST-2. The proposed method KARMA consistently achieves competitive or superior performance in most settings.

Table 1: Overall Performance on IMDb and SST-2 for KARMA

| Dataset | Metric | Acc | | | | | | F1 | | | | | |
|---|---|---|---|---|---|---|---|---|---|---|---|---|---|
| | | Ori | Ret | Fin | GA | KGA | KARMA | Ori | Ret | Fin | GA | KGA | KARMA |
| IMDb | $\mathcal{D}_f$ | 98.65 | 91.93 | **90.13** | 98.21 | 96.86 | 96.41 | 98.65 | 91.93 | **90.09** | 98.20 | 96.85 | 96.40 |
| | $\mathcal{D}_r$ | 99.12 | 98.25 | 98.13 | 98.97 | 99.89 | **98.28** | 99.12 | 98.25 | 98.13 | 98.97 | 99.89 | **98.28** |
| | $\mathcal{D}_{kt}$ | 91.48 | 93.72 | **93.27** | 91.48 | **93.27** | **93.27** | 91.46 | 93.72 | 93.26 | 91.46 | **93.27** | 93.26 |
| | $\mathcal{D}_{kr}$ | 99.55 | 97.76 | 96.86 | 99.55 | 100.00 | **96.41** | 99.55 | 97.76 | 96.86 | 99.55 | **100.00** | 97.76 |
| | $\mathcal{D}_t$ | 92.28 | 92.48 | 90.60 | 92.32 | **92.84** | 91.88 | 92.27 | 92.47 | 90.59 | **92.31** | 92.84 | 91.87 |
| SST-2 | $\mathcal{D}_f$ | 96.20 | 93.06 | **93.39** | 95.87 | 99.83 | 95.21 | 96.14 | 92.92 | **93.30** | 95.81 | 99.83 | 95.14 |
| | $\mathcal{D}_r$ | 97.20 | 97.06 | 96.76 | **97.13** | 99.96 | 98.86 | 97.16 | 97.02 | 96.72 | **97.10** | 99.96 | 98.84 |
| | $\mathcal{D}_{kt}$ | 97.19 | 97.85 | 97.52 | 97.02 | **58.02** | 98.51 | 96.22 | 97.06 | 96.67 | 96.01 | **57.43** | 97.97 |
| | $\mathcal{D}_{kr}$ | 97.69 | 97.19 | 95.37 | **97.69** | 100.00 | 98.84 | 97.66 | 97.15 | 95.32 | **97.66** | 100.00 | 98.83 |
| | $\mathcal{D}_t$ | 94.46 | 94.37 | 93.94 | **94.49** | 94.52 | 94.85 | 94.42 | 94.30 | 93.89 | **94.45** | 94.45 | 94.80 |
| | Size (Mb) | - | - | 417 | 417 | 1251 | **2.31** | | | | | | |

For $\mathcal{D}_r$, KARMA achieves comparable performance to Retrain, and even outperforms other baselines, demonstrating the effectiveness on preserving irrelevant knowledge. Regarding $\mathcal{D}_f$, KARMA

Table 2: Performance comparison of KARMA and KARMA$_{KO}$ on IMDb and SST-2.

| Method | | | IMDb | | | | | SST-2 | | | | |
|---|---|---|---|---|---|---|---|---|---|---|---|---|
| | | | $\mathcal{D}_f$ | $\mathcal{D}_r$ | $\mathcal{D}_{kt}$ | $\mathcal{D}_{kr}$ | $\mathcal{D}_t$ | $\mathcal{D}_f$ | $\mathcal{D}_r$ | $\mathcal{D}_{kt}$ | $\mathcal{D}_{kr}$ | $\mathcal{D}_t$ |
| KARMA | Acc | | 95.52 | 98.26 | 90.58 | 96.41 | 91.88 | 94.38 | 96.54 | 97.36 | 97.52 | 95.92 |
| | F1 | | 95.51 | 90.57 | 90.58 | 96.41 | 91.88 | 94.34 | 96.51 | 96.49 | 97.49 | 95.89 |
| KARMA$_{KO}$ | Acc | | 95.96 | 97.91 | 92.38 | 95.96 | 91.40 | 92.40 | 93.16 | 95.87 | 92.40 | 92.11 |
| | F1 | | 95.96 | 97.90 | 91.93 | 96.85 | 91.36 | 92.36 | 92.46 | 92.49 | 92.54 | 91.76 |

also outperforms GA and KGA, while still maintaining better $\mathcal{D}_r$. KARMA also performs well on $\mathcal{D}_t$ across both datasets, indicating its strong generalizability. KARMA performs slightly better than retrain on both $\mathcal{D}_{kt}$ and $\mathcal{D}_{kr}$ in IMDB, and similar in SST-2, suggesting that it better eliminates related knowledge in unseen subsets while preserving semantic information. In terms of modified data size, KARMA is only 2.31 MB, compared to 417 MB for Fin and GA, and 1251 MB for KGA (since it requires full fine-tuning of multiple models). Overall, KARMA achieves comparable unlearning effectiveness with fewer parameter updates and fine-tuning steps, demonstrating the method's ability to effectively constrain the fine-tuning scope, achieving an effective trade-off between forgetting target knowledge and preserving retained information.

**Privacy analysis.** To assess the privacy-preserving capabilities of our method, we conduct Membership Inference Attacks (MIA). The $\Delta$ASR reflects the privacy protection capability of a method under MIA attacks. The $\Delta$AUC quantifies the attacker's ability to distinguish members from non-members, with 0.5 indicating random guessing. As shown in Figure 3 and Figure 4, KARMA outperforms the retraining strategy on both metrics, indicating its targeted effectiveness in removing semantic traces of the forgotten samples. Moreover, KARMA with keyword-aware only strategy (KARMA$_{KO}$) achieves even better results, which may be attributed to the greater semantic shift introduced by fine-tuning on pseudo samples. This indicates that KARMA can achieve security performance comparable to or even surpassing that of retraining. In particular, for privacy-sensitive scenarios, KARMA$_{KO}$ makes it difficult for malicious attackers to extract information about the forgotten samples through model confidence scores.

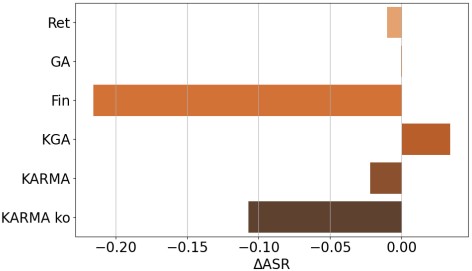

Figure 3: Comparison of $\Delta$ASR. The x-axis represents the performance difference in the ASR of MIA between each baseline and the $\theta_p$.

Figure 4: Comparison of $\Delta$ACU. The x-axis represents the difference between the AUC values of each baseline and 0.5.

**Ablations.**

(1)As shown in Table 2, we compare the performance of full-sample (KARMA) and key-aware-only (KARMA$_{KO}$) modes of KARMA. The comparison shows that while KARMA$_{KO}$ effectively reduces retention of forgotten information, it suffers from decreased accuracy on remain and clean sets and over-removal indicated by worse DKR performance. This is because pseudo-samples generated from keywords cannot fully capture the original data's semantic richness, introducing noise that harms generalization. In contrast, KARMA's fine-tuning on real forgotten data achieves more precise updates, better preserving overall model performance while effectively forgetting sensitive information. Despite its limitations, KARMA$_{KO}$ offers a practical and privacy-friendly alternative when access to original data is restricted, making a reasonable trade-off between privacy and unlearning effectiveness.

(2) To investigate the impact of historical information on model performance, we first analyze the forgetting behavior under different values of the hyperparameter $\alpha$. We vary $\alpha$ from 0.0 (no use of historical knowledge) to 1.0 (only relying on historical knowledge), and measure the sum of absolute differences in accuracy on $\mathcal{D}_f$, $\mathcal{D}_r$, and $\mathcal{D}_t$ compared to the baseline. As shown in Figure 5, effective unlearning of the target knowledge can only be achieved when sufficient historical information is preserved (metric-wise results can be found in Table 4).

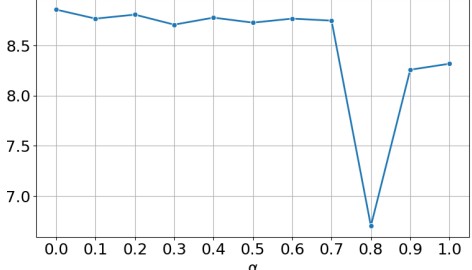
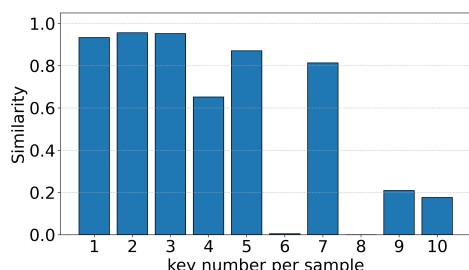

Figure 5: Performance with different $\alpha$. The y-axis shows the absolute performance difference on $\mathcal{D}_f$ between the model under different $\alpha$ values and the retrained model.

Figure 6: Performance with different keyword number. Similarity is defined as the normalized composite difference between the model and the retrained model on $\mathcal{D}_f$, $\mathcal{D}_r$, and $\mathcal{D}t$.

(3) We further examine how the number of selected keywords influences forgetting performance. Based on keyword proportions derived from our observations, we test the model with 1 to 10 keywords. Figure 6 illustrates the corresponding changes in performance ($\mathcal{D}_f$, $\mathcal{D}_r$, $\mathcal{D}_t$ accuracy differences from the baseline) under different keyword counts. Interestingly, a smaller, semantically focused keyword set tends to yield better forgetting performance, while an overly large set may degrade it by increasing the chance of keyword overlap with retained samples, thus weakening the anchoring effect and causing unintended spillover (see metric-wise results and keyword extraction study in Table 5 and Table 6).

(4) Table 3 shows the performance of the proposed KARMA under different size of $\mathcal{D}_f$. Although performance declines as the proportion of forgotten samples increases, KARMA maintains a consistent level of effectiveness. When forgetting no more than 10% of samples, KARMA's performance remains close to that of full retraining, demonstrating its robustness in balancing forgetting and retention.

Table 3: Performance with different unlearning rate $\beta$ (%)

| Metrics | Accuracy | | | | | |
| | $\beta = 1\%$ | $\beta = 5\%$ | $\beta = 10\%$ | $\beta = 15\%$ | $\beta = 20\%$ | $\beta = 30\%$ |
|---|---|---|---|---|---|---|
| $\mathcal{D}_f$ | 92.38 | 95.46 | 94.71 | 94.99 | 93.29 | 88.83 |
| $\mathcal{D}_t$ | 92.32 | 92.12 | 92.04 | 91.84 | 91.76 | 86.47 |

## 6  CONCLUSION

In this paper, we propose KARMA, a lightweight and efficient unlearning method that selectively modifies representation space to achieve unlearning. The design of KARMA ensures stable model behavior and preserves the performance on remaining samples by constraining updates within a bounded space. We theoretically demonstrate that the proposed regularization strategy effectively controls semantic drift and maintains robustness on retained data. Experimental results show that KARMA achieves comparable forgetting performance with a huge reduction in parameter updates and minimal degradation on remaining samples. In addition, KARMA effectively defends against MIA under both unlearning scenarios, demonstrating strong privacy-preserving capabilities.

## 7 ETHICS STATEMENT

This work adheres to the ICLR Code of Ethics. In this study, no human subjects or animal experimentation was involved. All datasets used were sourced in compliance with relevant usage guidelines, ensuring no violation of privacy. We have taken care to avoid any biases or discriminatory outcomes in our research process. No personally identifiable information was used, and no experiments were conducted that could raise privacy or security concerns. We are committed to maintaining transparency and integrity throughout the research process.

## 8 REPRODUCIBILITY STATEMENT

We have made every effort to ensure that the results presented in this paper are reproducible. All code and datasets have been made publicly available in an anonymous repository to facilitate replication and verification. The experimental setup, including training steps, model configurations, and hardware details, is described in detail in the paper and appendix. We have also provided a full description of keyword selection and pseudo-sample generation, to assist others in reproducing our experiments.

Additionally, public dataset like IMDb and SST-2 are publicly available, ensuring consistent and reproducible evaluation results.

We believe these measures will enable other researchers to reproduce our work and further advance the field.

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

## 9 APPENDIX

### 9.1 EXPERIMENTAL IMPLEMENTATION

All experiments were conducted on a server running CentOS Linux 7, equipped with a single NVIDIA RTX 4090 GPU (24GB). The implementation is based on the PyTorch framework. For reproducibility, the exact versions of software libraries and dependencies can be found in our open-source code repository: https://anonymous.4open.science/r/KARMA-4501. The following ablation studies are conducted using hyperparameters $lr$ = 2e-2, $\gamma$ = 0.8, and epoch = 1 for KARMA. For Ablation (3), $\alpha$ is set to the default value of 0.8.

Details of baselines are as follows:

**Retrain (Ret).** After removing $\mathcal{D}_f$ from $\mathcal{D}$, we fine-tuned the pre-trained model on the remaining samples to obtain the retrained model. The learning rate of retraining is set to be $2 \times 10^{-5}$ and run for 3 epochs. This approach acts as the baseline for all other models, and it is required that the quantitative metrics of other methods be as similar as possible to this method.

**Fine-tuning (Fin).** After removing $\mathcal{D}_f$ from $\mathcal{D}$, we continued to fine-tune the whole model parameters on the remaining samples. The learning rate of fine-tuning is set to $1 \times 10^{-4}$ and run for 3 epochs.

**Gradient Ascent (GA).** The strategy of gradient ascent is widely used in previous works to achieve unlearning Jang et al. (2023); Yuanshun et al. (2023). Considering the differences in scenarios and objectives, we have implemented a basic version of the gradient ascent method as an alternative. We employed a gradient ascent strategy on $\mathcal{D}_f$, using a smaller learning rate for fine-tuning. The learning rate of GA is set to $1 \times 10^{-6}$ and run for 3 epochs.

**KGA.** KGA achieves an effect similar to KARMA by aligning the model with a knowledge-retaining counterpart trained on the remain dataset. Most experimental settings follow the configurations in Reference Wang et al. (2023a), with the learning rate adjusted to $2 \times 10^{-5}$ and the auxiliary dataset $\mathcal{D}_n$ set to a size of 100, aiming to achieve reasonable performance on IMDb and SST-2.

### 9.2 METRIC-WISE PERFORMANCE UNDER VARYING $\alpha$

As shown in Table 4, the model achieves effective forgetting of the target data without overfitting when $\alpha$ is set between 0.7 and 0.8.

Table 4: Metric-wise performance under varying $\alpha$.

| $\alpha$ | $\mathcal{D}_f$ | $\mathcal{D}_r$ | $\mathcal{D}_{kt}$ | $\mathcal{D}_{kr}$ | $\mathcal{D}_t$ |
|---|---|---|---|---|---|
| 0.0 | 99.10 | 99.39 | 95.07 | 98.65 | 93.32 |
| 0.1 | 98.65 | 99.39 | 95.07 | 98.65 | 93.32 |
| 0.2 | 99.10 | 99.36 | 94.62 | 98.65 | 93.36 |
| 0.3 | 99.10 | 99.34 | 94.17 | 98.65 | 93.44 |
| 0.4 | 99.10 | 99.30 | 94.62 | 98.65 | 93.36 |
| 0.5 | 98.65 | 99.08 | 93.72 | 98.65 | 92.60 |
| 0.6 | 98.65 | 99.12 | 94.17 | 98.21 | 92.80 |
| 0.7 | 95.96 | 97.86 | 90.58 | 97.31 | 91.04 |
| 0.8 | 97.76 | 98.94 | 92.83 | 98.65 | 92.60 |
| 0.9 | 98.65 | 99.20 | 94.17 | 98.65 | 92.92 |
| 1.0 | 98.21 | 99.00 | 93.72 | 98.65 | 92.52 |

### 9.3 METRIC-WISE PERFORMANCE UNDER VARYING NUMBERS OF KEYWORDS

As shown in Table 5, a smaller keyword set helps retain more knowledge on $\mathcal{D}_r$. In contrast, using too many keywords may cause excessive forgetting, leading to poor performance on both $\mathcal{D}_f$ and $\mathcal{D}_r$.

Table 5: Metric-wise performance under varying numbers of keywords.

| Keyword Num | $\mathcal{D}_f$ | $\mathcal{D}_r$ | $\mathcal{D}_{kt}$ | $\mathcal{D}_{kr}$ | $\mathcal{D}_t$ |
|---|---|---|---|---|---|
| 1 | 99.10 | 99.39 | 95.07 | 98.65 | 93.32 |
| 2 | 96.86 | 98.63 | 91.93 | 97.31 | 92.24 |
| 3 | 72.65 | 72.56 | 73.54 | 67.26 | 70.67 |
| 4 | 61.43 | 60.68 | 53.36 | 54.71 | 60.38 |
| 5 | 52.47 | 49.47 | 45.74 | 46.19 | 50.78 |
| 6 | 89.69 | 90.20 | 84.75 | 85.20 | 86.51 |
| 7 | 59.19 | 61.00 | 53.36 | 57.40 | 60.06 |
| 8 | 67.71 | 73.05 | 60.54 | 65.47 | 72.07 |
| 9 | 82.06 | 88.42 | 87.44 | 89.69 | 84.15 |
| 10 | 85.65 | 89.46 | 87.44 | 89.24 | 86.03 |

## 9.4 METRIC-WISE PERFORMANCE ACROSS VARIOUS KEYWORD SELECTION METHODS

We evaluate three keyword selection strategies: GradInput, TF-IDF, and Random. GradInput estimates token importance by computing the gradient of the loss with respect to input embeddings, following saliency-based interpretability techniques Shrikumar et al. (2017). TF-IDF is a classical

Table 6: Metric-wise performance across various keyword selection methods.

| **Metric** | GradInput | Tf-idf | Random | Ours |
|---|---|---|---|---|
| $\mathcal{D}_f$ | 96.41 | 90.58 | 92.38 | 96.86 |
| $\mathcal{D}_r$ | 98.36 | 95.24 | 94.90 | 98.63 |
| $\mathcal{D}_{kt}$ | 91.93 | 89.69 | 89.24 | 91.93 |
| $\mathcal{D}_{kr}$ | 97.76 | 94.62 | 93.72 | 97.31 |
| $\mathcal{D}_t$ | 92.00 | 89.68 | 89.36 | 92.24 |

statistical method that assigns importance scores based on a token's term frequency and its inverse document frequency across the corpus Grootendorst (2022). Random serves as a baseline by assigning each token a random importance score sampled uniformly from the range $[0, 1)$. For all methods, we retain semantically meaningful tokens, enumerate all $n$-gram combinations, compute their mean importance scores, and rank them by absolute value. The top-$k$ $n$-grams are selected as keywords ($n = 2$, $k = 2$). As shown in Table 6, we apply grid search to optimize hyperparameters for each method. While all approaches support effective forgetting under the 2-gram top-2 setting, our method better preserves performance on remaining data.

## 9.5 THE USE OF LARGE LANGUAGE MODELS (LLMS)

We confirm that large language models (LLMs) were used in the preparation of this manuscript. Specifically, LLMs were employed only to aid or polish writing, such as improving grammar, style, and readability. They were not used for generating research ideas, designing experiments, analyzing results, or producing any scientific content. All substantive contributions are solely from the authors.