# OpenReview forum: "KARMA: Keyword-Aware Representation Modification for Efficient and Robust Model Amnesiac Unlearning"
_ICLR.cc/2026/Conference — Submitted to ICLR 2026_

### Official Review · Reviewer_X7Ms · 2025-10-29

**Soundness:** 1
**Presentation:** 2
**Contribution:** 2
**Rating:** 2
**Confidence:** 5

**Summary:**

The paper proposes KARMA, a framework for machine unlearning in pre-trained Language Models. The method is based on the key insight that fine-tuning a single layer can match the effectiveness of full-model fine-tuning. KARMA unlearns by: Identifying semantically critical tokens via a Fisher scoring mechanism, Perturbing only their embedding representations, and Constraining updates within a bounded spherical region to preserve utility and model stability. To further improve privacy and efficiency, KARMA uses a keyword-driven pseudo-sample generation method.

**Strengths:**

By updating only one layer’s embeddings and using token-level precision, KARMA achieves near-retraining results with 99.5% fewer parameter updates.

**Weaknesses:**

1. The experiments are limited to relatively simple datasets such as IMDb and Sentiment. These are not representative of real-world unlearning challenges. It would strengthen the paper to evaluate on more diverse and difficult benchmarks (e.g., ToFU, WMDP) and to include utility assessments using broader metrics such as MMLU or World Facts.

2. Please include comparisons against other parameter-efficient unlearning methods, such as LoRA-based and embedding corruption approaches [1], to better position the proposed method within the efficiency–performance trade-off landscape.

3. The paper should provide a more detailed analysis of computational cost and storage requirements relative to other unlearning techniques including the other parameter-efficient unlearning methods. This would clarify the claimed efficiency advantages.

4. Since the method claims to preserve model utility, additional evaluations should be included to substantiate this claim. Comparisons with other unlearning methods that explicitly address utility preservation would be particularly informative.

5. The paper argues that LoRA is not well-suited for unlearning because it limits parameter updates. However, prior work [2–5] has demonstrated that LoRA-based unlearning can be effective. The authors should clarify how their claim aligns or conflicts with these findings.

6. The paper may suffers from that that linguistically important yet semantically uninformative tokens tend to have low per-token loss due to frequent exposure during pretraining [6]. It would be valuable to analyze or visualize the quality of the selected tokens.

7. The proposed method may be vulnerable to relearning or re-identification attacks. Experiments evaluating robustness under such attack settings would improve the paper’s completeness and practical relevance.

[1] Liu, Chris, et al. "Large language model unlearning via embedding-corrupted prompts." Advances in Neural Information Processing Systems 37 (2024): 118198-118266.

[2] Gao, Chongyang, et al. "On Large Language Model Continual Unlearning." The Thirteenth International Conference on Learning Representations.

[3] Cha, Sungmin, et al. "Towards robust and parameter-efficient knowledge unlearning for llms." arXiv preprint arXiv:2408.06621 (2024).

[4] Kim, Yejin, et al. "Improving Fisher Information Estimation and Efficiency for LoRA-based LLM Unlearning." Second Conference on Language Modeling.

[5] Liu, Zheyuan, et al. "Towards Safer Large Language Models through Machine Unlearning." Findings of the Association for Computational Linguistics ACL 2024. 2024.

[6] Duan, Michael, et al. "Do Membership Inference Attacks Work on Large Language Models?." First Conference on Language Modeling.

**Questions:**

Please refer to the Weaknesses section for detailed discussion.

---

### Official Review · Reviewer_Lec8 · 2025-11-01

**Soundness:** 3
**Presentation:** 3
**Contribution:** 3
**Rating:** 4
**Confidence:** 4

**Summary:**

This paper introduces KARMA (Keyword-Aware Representation Modification for Model Amnesiac Unlearning), a novel and efficient framework designed to address the challenges of model unlearning in large-scale Pre-trained Language Models (PLMs), namely the high computational cost and the imbalance between forgetting effectiveness and utility preservation.

**Strengths:**

- Exceptional Efficiency: By limiting parameter updates exclusively to a small subset of the embedding layer's parameters, KARMA drastically reduces the computational overhead associated with unlearning. This is a crucial practical contribution given the massive scale of modern PLMs.

- Targeted Knowledge Localization: The approach of identifying and modifying keyword embeddings based on a Fisher scoring mechanism is a theoretically sound and highly targeted method. This contrasts favorably with methods that apply general, unlocalized parameter updates across all layers.

- Clear Utility Preservation Mechanism: The use of a bounded spherical region to constrain parameter changes provides an explicit, interpretable mechanism for preserving the original model's stability and mitigating the risk of catastrophic forgetting.

**Weaknesses:**

1. Application to Generative Tasks: The current experimental validation is primarily focused on classification-style tasks (e.g., measuring perplexity, discrimination). It remains unclear how effectively the KARMA strategy—which modifies input representations—applies to more complex and high-stakes generation tasks, where subtle changes in embeddings can lead to significant shifts in fluency, coherence, or safety. The authors should discuss the feasibility and potential challenges of applying KARMA to unlearn harmful or specific generative knowledge.

2. Model and Scale Generalization: The experiments are currently limited to the BERT-base-uncased model. To confirm the generalizability and practical value of KARMA, the authors should extend the evaluation to:

Different Architectures: Test on other popular PLM architectures (e.g., RoBERTa, and dedicated decoder-only models).

Larger Models: Demonstrate the efficiency and effectiveness of the strategy on significantly larger models (e.g., 7B or larger) to validate its scalability, which is a key claim of the paper.

3. Dataset Timeliness and Challenge: The datasets used in the current experimental setup appear to be older. It is recommended to include experiments on more recent and challenging datasets  to ensure the proposed method's robustness against modern, complex data distributions and real-world unlearning scenarios.

**Questions:**

see weakness

---

### Official Review · Reviewer_Exam · 2025-11-01

**Soundness:** 2
**Presentation:** 1
**Contribution:** 2
**Rating:** 2
**Confidence:** 4

**Summary:**

This paper proposes KARMA, a method for LLM unlearning. Specifically, the authors first identify the important keywords for each sample in the forget set, based on the Fisher scoring. They then only fine-tune the embedding layer of the tokens that correspond to these keywords. On the forget set, they minimize the cross-entropy with a different label sampled from the dataset. On the retain set, they minimize the KL with the original model. The authors conduct experiments on two sentiment analysis datasets to verify the effectiveness of the proposed method.

**Strengths:**

1. The paper includes theoretical analyses to support the proposed method.
2. The proposed method is efficient.

**Weaknesses:**

1. The method assumes access to the original pre-trained model before fine-tuning, which should not be available for an unlearning method. In widely used LLM unlearning benchmarks, if the goal is to forget knowledge learned during fine-tuning, then only the fine-tuned model should be given [1]. If the goal is to forget knowledge learned during pre-training, then only the pre-trained model should be given [2].
2. The performance of the proposed method is not good compared to the baselines. Particularly, in Table 1, the proposed method is worse than the Fin baseline.
3. The paper only evaluates on two datasets with classification tasks on sentiment analysis, which is a relatively toy setting and not applicable in real-world scenarios. Specifically, the authors should evaluate on more popular benchmarks that require generation or involve forgetting real-world knowledge from LLMs [1-3].
4. The presentation of the paper is not clear. For example, equation 2 is an update rule, not an optimization objective. Equation 3 is very confusing; $\mathbb{E}$ should represent the expectation, not cross-entropy.


[1] Maini et al., TOFU: A Task of Fictitious Unlearning for LLMs.

[2] Shi et al., MUSE: Machine Unlearning Six-Way Evaluation for Language Models.

[3] Li et al., The WMDP Benchmark: Measuring and Reducing Malicious Use With Unlearning.

**Questions:**

Please see weaknesses.

---

### Official Review · Reviewer_N1aj · 2025-11-02

**Soundness:** 3
**Presentation:** 3
**Contribution:** 2
**Rating:** 2
**Confidence:** 4

**Summary:**

The paper presents KARMA, a lightweight and targeted method for machine unlearning in language models. It selectively fine-tunes only the embeddings of high-influence “keywords,” identified through a Fisher-scoring criterion, to forget specific training samples while minimizing the change made to the rest of the model. A bounded regularization step constrains updates to prevent semantic drift, and a privacy-preserving variant performs unlearning using pseudo-samples built from keywords alone. Experiments with BERT-base on sentiment tasks show KARMA achieves unlearning comparable to retraining with over 99% fewer parameter updates and improved resistance to membership-inference attacks.

**Strengths:**

1. Fisher-based keyword selection is an interesting way to select what parameters should be modified in the model, and it makes sense that training fewer parameters will hurt less of the retain set performance.
2. The ablations highlight that the method works as intended and the regularization hyperparameter is meaningful.

**Weaknesses:**

1. The experimental setting is way too small to be realistic. I generally refrain from making this type of comment in a review, since it's always easy to ask for larger models / more datasets. But in this case, BERT-base and sentiment tasks are dramatically different from the cases where we would be interested in effectively achieving unlearning. At the very least it would make sense to look at smaller-scale / well-structured PII-related settings and larger models. In its current state, the experiment setting is so far from the relevant one that they give me essentially no prior on whether or not this will work in a setting where we really need to unlearn data.

2. Fisher importance is an unstable signal, because gradients can vary dramatically even just based on random seeds. I am not sure that these token-level sensitivities are robust indicators of semantic importance. This relates to my above comment as well, because I think it is especially easy to identify sentiment-related keywords (e.g., good or bad) but the same is not always true for cases where we want to perform unlearning.

3. The theoretical analysis is not very meaningful. There are established theoretical notions (via differential privacy) that are entirely ignored, and I don't think that the smoothness conditions hold strongly enough in modern-day models for this type of approach to make sense. The authors know this and acknowledge it, but I still think it is worth highlighting as a weakness.

4. Because the empirical setting is so far from the modern day LM setting, a number of baselines cannot even be evaluated. This is an important point, because methods that operate via in-context learning don't make any changes to the parameters and should sort of vacuously work even better than their method according to their theoretical analysis. In general, it's hard to contextualize this work in the vast body of unlearning work due to the size + scale of the experiments.

**Questions:**

1. Why did you not run your unlearning method on any established unlearning benchmarks such as TOFU and MUSE? Are there reasons to believe that your algorithm will work in true settings where we care about unlearning (e.g., large models + PII)?

---

### Meta-Review · Area_Chair_Zp12 · 2025-12-09

**Summary:**

The evaluation is too small and unrepresentative, relying on BERT-base sentiment datasets with mixed results against simple baselines, and in one table underperforming Fin. The method’s assumptions and rationale are shaky, including access to the original pre-trained model, unstable Fisher-based keyword importance, and weak or confusing theory and objectives with unclear equation definitions. Key evaluation gaps remain, such as tests on modern unlearning benchmarks and generative tasks, comparisons to parameter-efficient baselines like LoRA and embedding-corruption, and thorough compute, storage, and utility-preservation analyses. Robustness and practicality are uncertain with limited evidence of generalization to larger models and diverse datasets.

**Reviewer Concerns:**

No response was given.

**Reviewer Scores:**

No response was given.

---

### Decision · Program_Chairs · 2026-01-26

Reject